# Establishment of a Mutant Library of *Fragaria nilgerrensis* Schlechtendal ex J. Gay via EMS Mutagenesis

**Shu Jiang, Mingqian Wang, Can Zhao, Yuchen Cui, Zhi Cai, Jun Zhao, Yang Zheng, Li Xue \* and Jiajun Lei \***

College of Horticulture, Shenyang Agricultural University, Shenyang 110866, China
* Correspondence: lixue@syau.edu.cn (L.X.); jiajunlei@syau.edu.cn (J.L.); Tel.: +86-13478198765 (J.L.); Fax: +86-024-88487146 (J.L.)

**Abstract:** The diploid wild strawberry *Fragaria nilgerrensis* Schlechtendal ex J. Gay mainly distributed in Southwest China has many excellent traits and a small genome. A high-quality genome of *F. nilgerrensis* is available, but functional genomic research remains scarce. In the present study, to promote functional genomic research of *F. nilgerrensis*, ethyl methane sulfonate (EMS) was used to mutagenize the apical meristems, and the appropriate EMS mutagenesis dosages were screened. After treatment of 1200 apical meristems with 0.6% EMS for 6 h, a mutant library consisting of 86 mutant individuals, which were characterized by 17 mutant types, with a mutation rate of 7.17% was established. The characteristics of mutants included changes in the color, shape, number and size of leaves, and the architecture of flower and plant. The obtained mutants were identified by morphological appearance, botanical indexes, chlorophyll, photosynthetic fluorescence assays, root-tip chromosome, and flow cytometry observation. These mutants can provide great resources for gene functional research and future breeding of *F. nilgerrensis*.

**Keywords:** strawberry; *Fragaria nilgerrensis*; EMS; mutant; apical meristem

## 1. Introduction

With a vast territory and diverse climate types, China is the main center of the genus *Fragaria* in the world and has the most abundant wild strawberry species [1]. There are 15 wild species, including 10 diploids and 5 tetraploids, in China, accounting for more than 50% of the 25 *Fragaria* species in the world [2–4]. Wild strawberry species have great breeding potential for mining excellent genes because of their valuable characteristics, such as good fruit quality, cold tolerance, disease resistance, and so on [5,6]. The wild diploid species *Fragaria nilgerrensis* is mainly distributed in Sichuan, Yunnan, Guizhou, Shaanxi, Hubei, Chongqing, and Hunan Provinces in China. It is a robust plant, trifoliate and obovate or subround with cuneate apex, sympodial runners, cymose inflorescence, bisexual flowers 1–2 cm in diameter, calyx clasping, white fruit with unique peach fragrance, strong disease resistance, and other characteristics [3,7]. At present, a new decaploid cultivar, 'Tokun', with large orange fruit with a strong peach flavor has been created by interspecific hybridization of *F. nilgerrensis* with octoploid strawberry cultivars [8], which has created a practical precedent for breeding cultivars using *F. nilgerrensis* resources. Genome sequencing and assembly of *F. nilgerrensis* have been finished [9] with a relatively small genome size (250 Mbp), which provides a valuable resource for gene function identification of this species. However, the functions of most of these excellent trait genes have not been clarified, and available tools and resources for functional genomics research in *F. nilgerrensis* are still very limited.

The construction of a high-density mutant library is one of the most efficient ways to identify novel genes, analyze mutant genes, and investigate the molecular basis for important agronomic traits [10–12]. In recent years, many mutants were obtained, and the related genes controlling important agronomic traits have been discovered from mutant

libraries [13–17]. For example, by screening mutants of a *Brassica campestris* L. ssp. *Chinensis* cultivar, a 40bp insertion that was responsible for the stay-green trait in the second exon of *Bra019346* of pakchoi mutant *nye* was identified [18]. In maize, ZmRE-1 retrotransposon was identified in the fifth exon of *Brachytic2* (*Br2*) of two dwarf mutants, sil1 and sil2, obtained in the space-flighted seeds, which indicated that the transposition of ZmRE-1 was probably correlated with the spaceflight [19]. Several methods have been used for plant mutant library construction, such as physical, chemical, and biological (insertional) approaches. Compared to physical mutagenesis with its difficult operation and insertional mutagenesis with its low efficiency [20,21], chemical mutagenesis is the most effective method, with the advantages of simple operation, cost-effectiveness, high mutation frequency, and fewer chromosomal aberrations [22].

EMS is a practical and extensively adopted mutagen in the construction of mutant populations. Progress has been made in the acquisition of mutants and the mining of important regulatory genes for related traits in strawberries. Research of EMS mutant suppressor of runnerless (srl) and reduced anthocyanins in petioles (rap) revealed that the DELLA protein of GA signaling and GST anthocyanin transporter influenced the runner formation and leaf and fruit coloration, respectively [23,24]. Identification of a GA-deficient mutant revealed that the *FvCYP714C2* gene plays an important role in gibberellin synthesis [25]. The conserved and novel roles of the miR164-CUC2 module in the development of deep serrated and deformed carpel in the leaves and flowers of woodland strawberry (*F. vesca*) EMS mutants have been analyzed [26]. In *F. vesca*, the gene *FveSEP3* and transcription factors GRAS, which influence leaf and fruit, development have been identified from EMS-mutated populations [27,28]. However, the construction of an *F. nilgerrensis* mutant library using EMS has not yet been reported. In this study, the apical meristem of *F. nilgerrensis* was treated with EMS, suitable mutagenesis conditions were screened, and a mutant library was established. The mutants were identified by morphology, physiology, cytology, and histology, which will lay the foundation for the genetic research and molecular breeding of *F. nilgerrensis*.

## 2. Materials and Methods

### 2.1. Plant Materials

The plants of *F. nilgerrensis* in our experiment with the accession code SN11-6 were collected from Yiliang County, Kunming City, Yunnan Province, China in 2020 and were planted in the open field of Shenyang Agricultural University.

The stolon meristems of *F. nilgerrensis* were used as explants and inoculated on MS medium to grow into plantlets. The plantlets were proliferated to be used for EMS mutagenesis on the medium MS + 0.2 mg·L$^{-1}$ 6-BA + 0.1 mg·L$^{-1}$ GA$_3$ supplemented with 30 g·L$^{-1}$ sucrose and 7 g·L$^{-1}$ agar.

### 2.2. EMS Dose Trials for Apical Meristems

The approximate 0.4–0.5 mm apical meristems peeled from in vitro plantlets of *F. nilgerrensis* were used for the preliminary test of a median lethal dose. EMS solutions (Macklin, Shanghai, China) at 0.0%, 0.3%, 0.6%, 0.9%, and 1.2% (*v/v*) were prepared, and the meristems were treated with each EMS solution for 3 h, 6 h, and 9 h with shaking at 120 rpm·min$^{-1}$ under 25 °C in the dark. Then they were rinsed three times with distilled water to prevent the adverse effects of residual reagents and transferred to the proliferation medium after being blotted dry with sterilized filter paper. Each treatment was replicated three times with 50 meristems.

Meristem survival was investigated on the 15th day after treatment. The survival rate of apical meristems is equal to the number of survivals divided by the total number of inoculated meristems multiplied by 100%. The treatment combinations that resulted in a median lethal dose were chosen as the optimal condition for subsequent experiments.

*2.3. Large-Scale EMS Treatment and Mutant Library Construction*

A total of 1200 apical meristems were treated with EMS dosage obtained from the above experiment, and the meristems without EMS treatment were used as the control. The treatment process was similar to the above preliminary test. Every survival meristem was propagated to more than 50 plantlets as one line. After rooting, the plantlets were transplanted into 72-hole plug trays and pots in turn. When the plantlets grew to about 10 cm in height with 5–8 leaves, they were transplanted to the open field for phenotypic observation.

The indexes of mutagenic plants determined to screen mutants included length and width of central leaflet, transect between tips of the two lateral leaflets, leaf curl index (LCI), leaf area, number of branch crowns, number of leaflets, number of leaves, plant height, crown diameter, leaf color, leaf shape, leaf thickness, number of flowers, flower diameters, number of petals, number of calyxes, number of stamens, fruit length, fruit weight, fruit color, and fruit fragrance according to the methods of Zhao et al. and Zhang et al. [29,30]. LCI is calculated as $(Lw - Ln)/Lw \times 100\%$. Lw is the widest width of the leaf when it is fully expanded, and Ln is the widest width of the leaf when it is naturally curling.

Through the determination of the botanical indexes, the mutagenic plantlets were classified according to the traits of plant, leaf, and flower, and the mutant library of *F. nilgerrensis* was established. Of the obtained mutants, 11 typical mutants were selected and named, including dark leaf (DL), yellow leaf (YL), curled leaf (CL), wrinkled leaf (WL), round leaf (RL), narrow leaf (NL), large leaf (LL), small leaf (SL), single leaf (SGL), dwarf plant (DP), and clumped plant (CP). They were abbreviated according to their phenotype mutation and used for data measurement in subsequent experiments compared to the wild-type plant (WT) as control.

*2.4. Measurement of Some Physiological Indexes of Mutants*

The L*, a*, and b* values of mutant and WT leaves were measured using a colorimeter (CR-400, Konica Minolta, Tokyo, Japan). The calculation method of total chlorophyll and chlorophyll a (Chla) and b (Chlb) contents referred to Tang et al. [31]. The absorbencies of the supernatants were spectrophotometrically quantified at 470 nm, 646 nm, and 663 nm, respectively. Their contents were calculated using the following formulas:

$$Chla \ (mg/g) = [(12.21OD663 - 2.81OD646) \times 8]/(leaf \ fresh \ weight \ (g) \times 1000);$$

$$Chlb \ (mg/g) = [(20.13OD646 - 7.32OD663) \times 8]/(leaf \ fresh \ weight \ (g) \times 1000);$$

$$Total \ Chl \ (mg/g) = [(20.29OD646 + 8OD470) \times 8]/(leaf \ fresh \ weight \ (g) \times 1000).$$

Photosynthesis-related indexes, including net photosynthetic rate (Pn, $\mu mol \ CO_2 \ m^{-2}s^{-1}$), stomatal conductance (Gs, $mmol \ H_2O \ m^{-2}s^{-1}$), intercellular carbon dioxide concentration (Ci, $\mu mol \ CO_2 \ mol^{-1}$), and transpiration rate (Tr, $mmol \ H_2O \ m^{-2}s^{-1}$), were measured using the LI-6400 photosynthesis instrument (LI-COR, Lincoln, Neb., USA). For chlorophyll fast-phase fluorescence kinetics, dark treatments were performed for 30 min before measuring fluorescence indexes, and the recording time course of the fluorescence signal was 2 s with a red light at 650 nm and a light source intensity of 3000 $\mu mol \cdot m^{-2}$. The measuring instrument was a Plant Efficiency Analyzer (Handy PEA, Hansatech, Norfolk, England). The analysis processing was performed by plotting Radar and OJIP (fluorescence transient curves: the fast phase is labeled as OJIP, where O is for origin, the first measured minimal level; J and I are intermediate levels; and P is the peak) charts with professional software provided by Lufthansa UK [32].

*2.5. Cytological Observation of Mutants*

The ploidy of mutants and WT was identified via chromosome observation of root-tip cells using the squash method according to Lei et al. [33] and flow cytometry (FACSCalibur, BD Biosciences, San Jose, CA, USA) according to Dolezel et al. [34].

*2.6. Data Analysis*

Each identification method was performed with three replicates for nine plants, and the average values were calculated. The SPSS statistical analysis software was used for ANOVA analysis, $p < 0.05$.

## 3. Results

*3.1. Identification of Optimal EMS Concentrations for Mutagenesis in F. nilgerrensis*

Four EMS doses and three treatment times were used to screen the optimal mutagenic conditions for the apical meristem of *F. nilgerrensis*. The results showed that there were significantly different survival rates of apical meristems among different EMS treatments (Table 1). At the same dose, the survival rate gradually decreased with the increase in treatment time. The survival rate also gradually decreased with the increase in treatment doses. These indicated that under the high concentration and long-term treatment of EMS, apical meristems would be severely impaired. With an EMS concentration of 1.2% for 9 h, the survival rate was 0.00%, indicating that this dose produced considerable toxicity to apical meristems. It was also found that at a low dose of 0.3% EMS for 3 h, the survival rate was relatively high (86.67%), yet this dose might generate a low mutation rate. In this experiment, the 50%-lethal dose of EMS was used as the treatment condition. The concentration of 0.6% EMS for 6 h produced a survival rate of 47%, which was close to the median fatality rate, and the mutant library was constructed under this treatment.

**Table 1.** The survival rate of apical meristem treated with EMS at different concentrations and duration in *F. nilgerrensis*.

| Treatment | CK | 0.30% | | | 0.60% | | | 0.90% | | | 1.20% | | |
|---|---|---|---|---|---|---|---|---|---|---|---|---|---|
| | | 3 h | 6 h | 9 h | 3 h | 6 h | 9 h | 3 h | 6 h | 9 h | 3 h | 6 h | 9 h |
| Survival rate (%) | 100.00 ± 0.00 a | 86.67 ± 4.73 b | 78.33 ± 5.86 c | 57.67 ± 2.52 e | 73.33 ± 4.04 d | 47.00 ± 2.00 f | 31.33 ± 2.08 g | 32.33 ± 2.08 g | 10.00 ± 1.00 h | 4.67 ± 1.53 ij | 6.00 ± 1.00 hi | 0.60 ± 1.15 j | 0.00 ± 0.00 j |

Note: Values represent mean ± SE; different lowercase letters indicate significant differences among treatments according to LSD test ($p < 0.05$).

*3.2. Large-Scale EMS Treatment and Mutant Library Construction of F. nilgerrensis*

The whole procedure spanned 9 to 10 months (Figure 1). A total of 1200 apical meristems of *F. nilgerrensis* were treated with 0.6% EMS for 6 h, and 611 survived, for a survival rate of 50.92%. Their botanical traits were investigated (Supplementary Table S1), and 86 (7.17%) individuals showed phenotypic variation.

The 86 mutants were classified into 3 categories and 17 types according to the characteristics of plant, leaf, and flower (Supplementary Figure S1; Table 2). Among these mutants, plant posture mutants, including dwarf plants and clumped plants, accounted for 2.33% of total mutants. The leaf mutant was the most frequent mutant, with a mutation rate of 4.00%, including leaf color, leaf shape, leaflet number, and leaf size (Supplementary Figure S2). Four new types of leaf color mutants were obtained, which were dark green, yellow-green, mottled yellow, and albino, with two, seven, five, and one mutant, respectively. For leaf shape mutants, five types were determined, including round leaf, curled leaf, wrinkled leaf, narrow leaf, and leaf vein bulge with four, two, one, one, and one mutant respectively. Moreover, one important mutant with one leaflet was distinct from the normal three leaflets. Flower mutation, with a 0.83% mutation rate, included flower size and petal number mutation (Supplementary Figure S3). Mutants with more than one mutagenic trait were observed. For example, one mutant showed the traits of small leaf and dwarf plant at the same time. Among those mutants, 11 typical mutants were screened (Figure 2), as described in Materials and Methods, and used to conduct subsequent experiments.

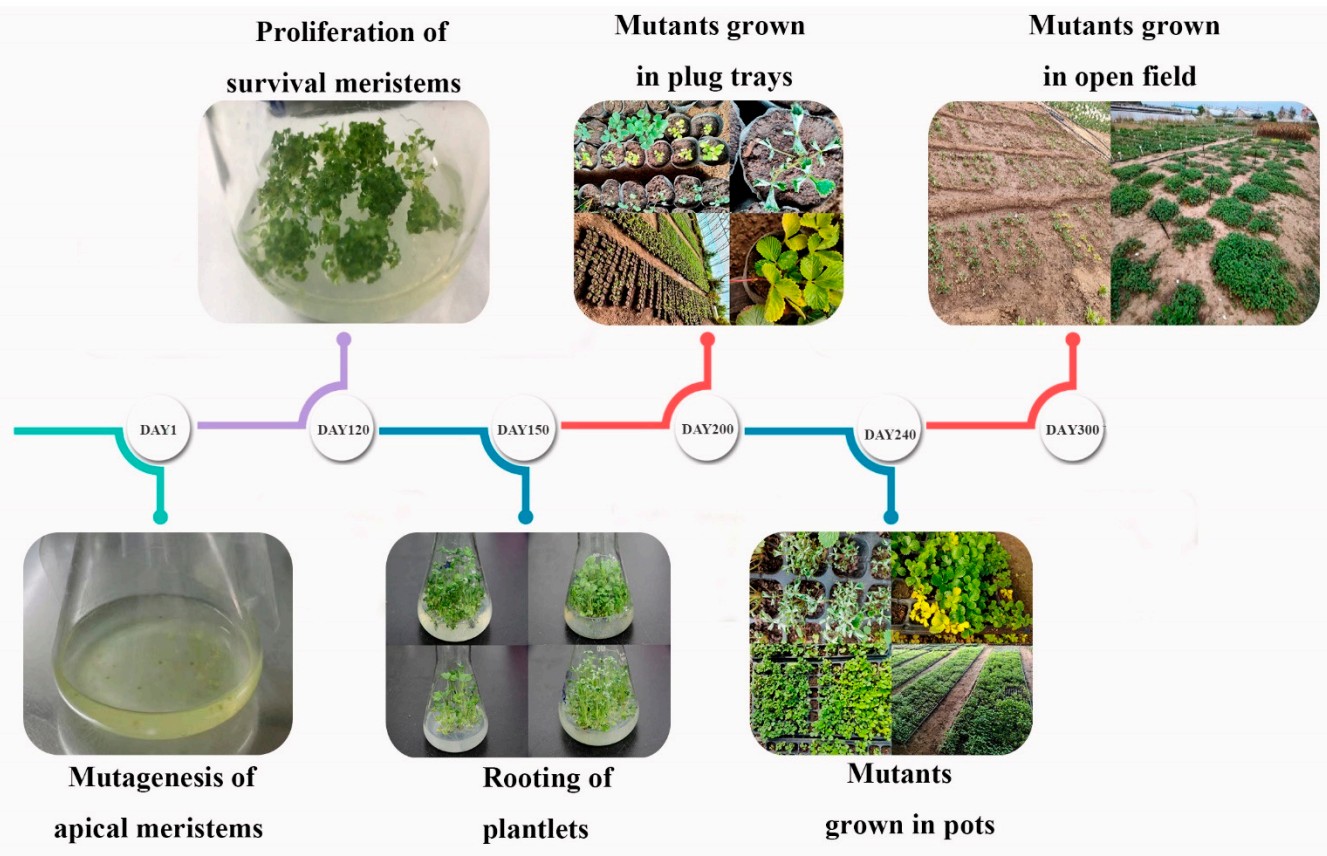

**Figure 1.** The establishment process of EMS mutant library in *F. nilgerrensis*.

**Table 2.** Statistics on EMS mutant types and their ratios in *F. nilgerrensis*.

| Mutant Categories | Mutant Types | No. of Mutants | Percentage of Mutation (%) |
|---|---|---|---|
| Plant posture | | 28 | 2.33 |
| | Dwarf plant | 26 | 2.17 |
| | Clumped plant | 2 | 0.17 |
| Leaf color and shape | | 48 | 4.00 |
| | Dark green leaf | 2 | 0.17 |
| | Yellow-green leaf | 7 | 0.58 |
| | Mottled yellow leaf | 5 | 0.42 |
| | Albino leaf | 1 | 0.08 |
| | Curled leaf | 2 | 0.17 |
| | Wrinkled leaf | 1 | 0.08 |
| | Round leaf | 4 | 0.33 |
| | Narrow leaf | 1 | 0.08 |
| | Leaf vein bulge | 1 | 0.08 |
| | Single leaflet | 1 | 0.08 |
| | Large leaf | 3 | 0.25 |
| | Small leaf | 20 | 1.67 |
| Flower characteristics | | 10 | 0.83 |
| | Large flower | 2 | 0.17 |
| | Small flower | 3 | 0.25 |
| | Increased petal number | 5 | 0.42 |
| Total | | 86 | 7.17 |

*3.3. Botanical Traits Analysis of Mutants in F. nilgerrensis*

3.3.1. Plant and Leaf

Plant and leaf morphology were significantly altered in EMS mutants (Figure 3A; Supplementary Table S1). Plant height, number of leaves, and branch crowns were mea-

sured for plant posture. As shown in Figure 3B, dwarf mutants with various degrees were screened, and all mutants were significantly shorter than WT, except for the LL mutant. For example, the DP mutant was extremely dwarf, with only 3.5 cm height, which was about 1/5 of WT. Mutants with compact and clustered types were screened. For example, the number of leaves and branch crowns of the CP mutant was significantly more than that of other mutants and WT (Figure 3C,D).

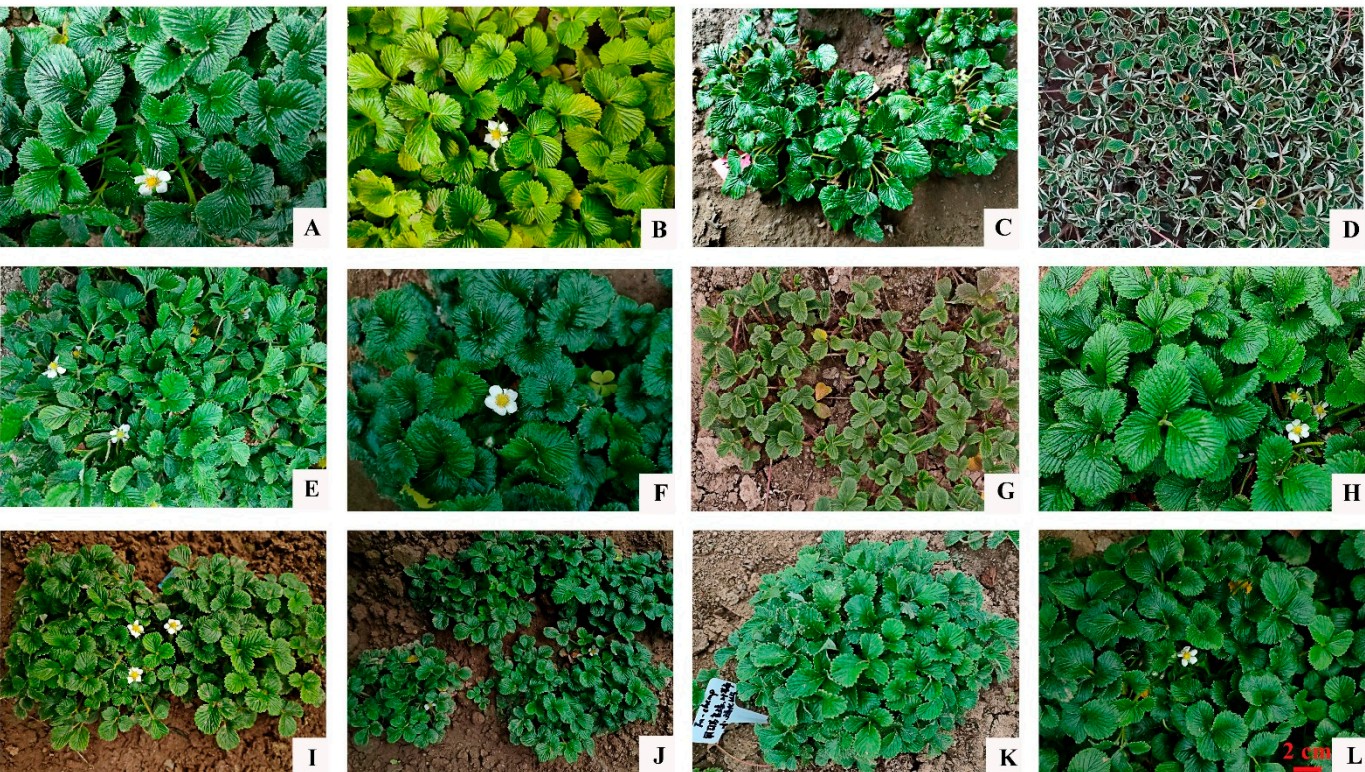

**Figure 2.** Plant and leaf phenotypes of 11 mutants in *F. nilgerrensis*. (**A**) DL: dark leaf; (**B**) YL: yellow leaf; (**C**) SGL: single leaf; (**D**) CL: curled leaf; (**E**) WL: wrinkled leaf; (**F**) RL: round leaf; (**G**) NL: narrow leaf; (**H**) LL: large leaf; (**I**) SL: small leaf; (**J**) DP: dwarf plant; (**K**) CP: clumped plant; (**L**) WT: wild-type plant. Scale bar = 2 cm.

To determine the leaf shape of mutants, the central leaflet length and width were measured. The central leaflet length/width ranged from 1.01 to 2.03 (Figure 3E). The value of the NL mutant with oblong leaves was 2.03, which was significantly higher than that of the RL and SGL mutants with round leaves. The LCI value ranged from 2–77%, and there was a significant difference between mutants (Figure 3F). The CL mutant showed the largest LCI value of 77%, compared to 8% of WT.

Mutants with possible ploidy variation can be identified by measuring the ratio of the two lateral leaflets' transect/central leaflet widths [30]. It can be seen from the measurement results of 11 mutants that the ratios of the WL mutant, RL mutant, and SGL mutant were reduced by 13.56%, 36.16%, and 43.50%, respectively, compared to WT (Table 3).

The mutants with different leaf sizes were obtained via mutagenesis. The leaf area of the central leaflet was evaluated (Figure 3G). The leaf area of the LL mutant was the highest (18.54 cm$^2$), while that of the SL mutant was the lowest (2.85 cm$^2$).

### 3.3.2. Flower

Of 11 mutants, ten mutants with different flower characteristics bloomed—the CL mutant did not bloom (Figure 4A; Supplementary Table S2). The WT had 10 calyxes and 21 stamens per flower and 3.3 flowers per inflorescence on average. However, the SGL mutant had 5 calyxes and 15 stamens and 8.67 flowers per inflorescence. The WL mutant

had significantly higher flower diameter (2.4 cm), petal number (7.67), calyx number (15.33), and stamen number (28.67) than WT (Figure 4B–E).

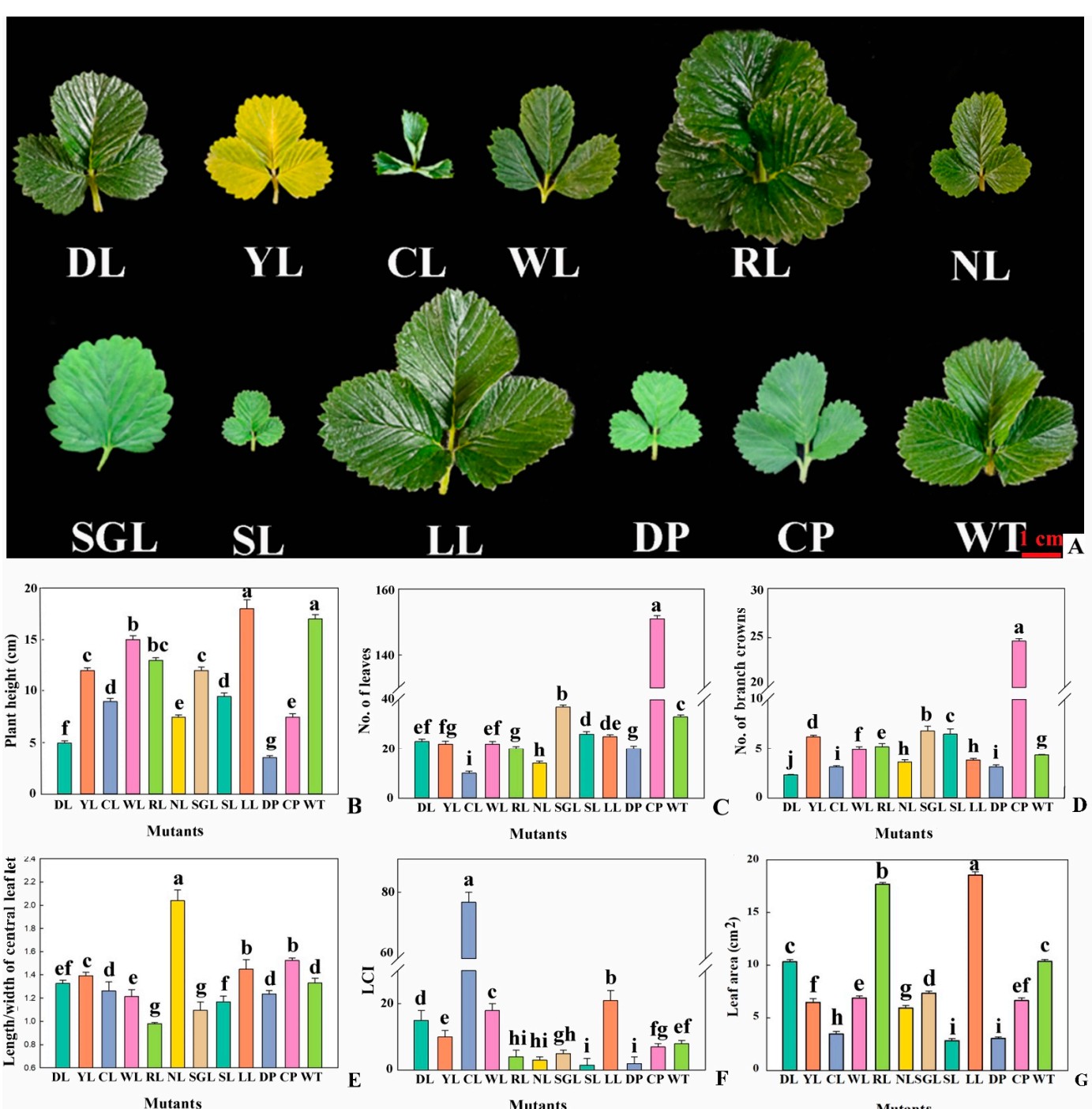

**Figure 3.** The plant and leaf traits of 11 mutants in *F. nilgerrensis*. (**A**) Leaves; (**B**) plant height; (**C**) No. of leaves; (**D**) No. of branch crowns; (**E**) leaf length/width; (**F**) LCI; (**G**) leaf area. Abbreviated letters of mutants were described in Figure 2. Bars indicate S.E. of means, and different lowercase letters indicate significant differences among treatments according to LSD test (*p* < 0.05). Scale bar = 1 cm.

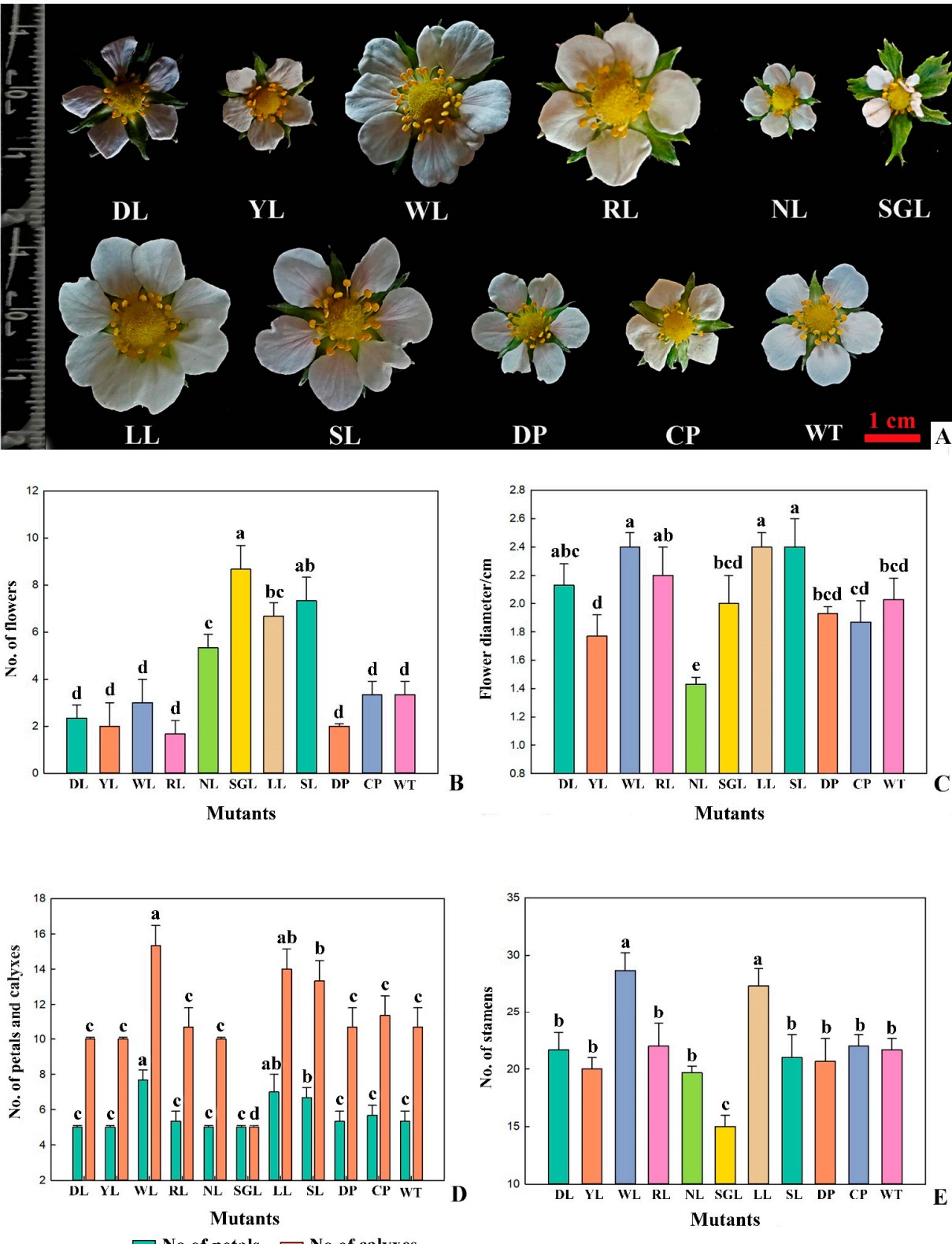

**Figure 4.** The flower traits of 10 mutants in *F. nilgerrensis*. (**A**) Flowers of 10 mutants; (**B**) No. of flowers; (**C**) flower diameter; (**D**) No. of petals and calyxes; (**E**) No. of stamens. Abbreviated letters of mutants were described in Figure 2. Bars indicate S.E. of means, and different lowercase letters indicate significant differences between treatments according to LSD test (*p* < 0.05). Scale bar = 1 cm.

**Table 3.** Statistics on the transect of the two lateral leaflets/width of the central leaflet in 11 mutants of *F. nilgerrensis*.

| Mutant Lines | The Transect of the Two Lateral Leaflets/Width of the Central Leaflet |
|:---:|:---:|
| DL | 1.72 ± 0.05 d |
| YL | 1.75 ± 0.04 d |
| CL | 3.68 ± 0.07 a |
| WL | 1.53 ± 0.05 e |
| RL | 1.13 ± 0.06 f |
| NL | 2.14 ± 0.06 b |
| SGL | 1.00 ± 0.01 g |
| SL | 1.78 ± 0.03 cd |
| LL | 1.84 ± 0.04 c |
| DP | 1.72 ± 0.03 d |
| CP | 1.78 ± 0.02 cd |
| WT | 1.77 ± 0.04 cd |

Note: Values represent mean ± SE; different lowercase letters indicate significant differences between treatments according to LSD test ($p < 0.05$).

### 3.3.3. Fruit

Among 11 mutants, nine mutants bore fruits except for CL and SGL mutants (Figure 5A). The fruits of the WL, NL, and DP mutants were deformed. The fruit of the RL mutant was nearly spherical, and the achenes were more evenly distributed on the fruit than in WT. The ratios of length/diameter of mutant fruits ranged from 0.76 to 1.04 and were not significantly different between mutants (Figure 5B,C; Supplementary Table S3). Compared to WT, the size and weight of the mutants' fruit were different. For example, the average fruit length and diameter of the RL mutant were 2.03 cm and 2.17 cm, respectively, which were 19.7% and 23% higher than that of WT, respectively. However, no changes were observed in fruit color and fragrance among mutants.

### 3.4. Measurement of Some Physiological Indexes of Mutants in F. nilgerrensis

### 3.4.1. The Values of L*, a*, and b*

To identify the leaf color variation of 11 mutants, L*, a*, and b* values of leaves were determined. The statistics showed that L* value ranged from 31.57 (black) to 63.23 (white), a* value ranged from −15.88 (yellow-green) to −7.7 (dark green), and b* value ranged from 11.9 (green) to 49.89 (yellow) (Supplementary Table S4). The YL mutant possessed yellow leaves with higher L* and b* values of 63.23 and 49.89, respectively, and a lower a* value of −18.55 compared to 38.27, 21.69, and −14.82, respectively, in WT. On the contrary, the DL mutant showed dark green leaves with the highest a* value and the lowest b* value. Based on the scatter plot of L*, a*, and b* of mutants and WT, mutants were divided into three groups (Figure 6A). The YL mutant formed a single group, which was classified as the yellow group. The SL, LL, WL, NL, SGL, DP, and CP mutants and WT formed the second group, which was classified as the green group. The CL, RL, and DL mutants formed the third group, which was classified as the dark green group. The a* value of the dark green group was the highest, followed by the green group and yellow group. The L* and b* values of the yellow group were the highest of the three groups. In summary, the L*, a*, and b* values were consistent with the mutant phenotypes.

### 3.4.2. The Contents of Chl, Chla, and Chlb

The Chl, Chla, Chlb, and Chla/Chlb values were measured to evaluate the changes in the pigment content of mutants. The results showed that mutants were significantly different compared to WT (Figure 6B). For example, the Chla, Chlb, and Chl values of the YL mutant were 0.43, 0.43, and 1.00 respectively, which were 1/5, 1/2, and 1/3 less, respectively, than those of WT; The Chla/Chlb value of the YL mutant was 1/3 less than that of WT.

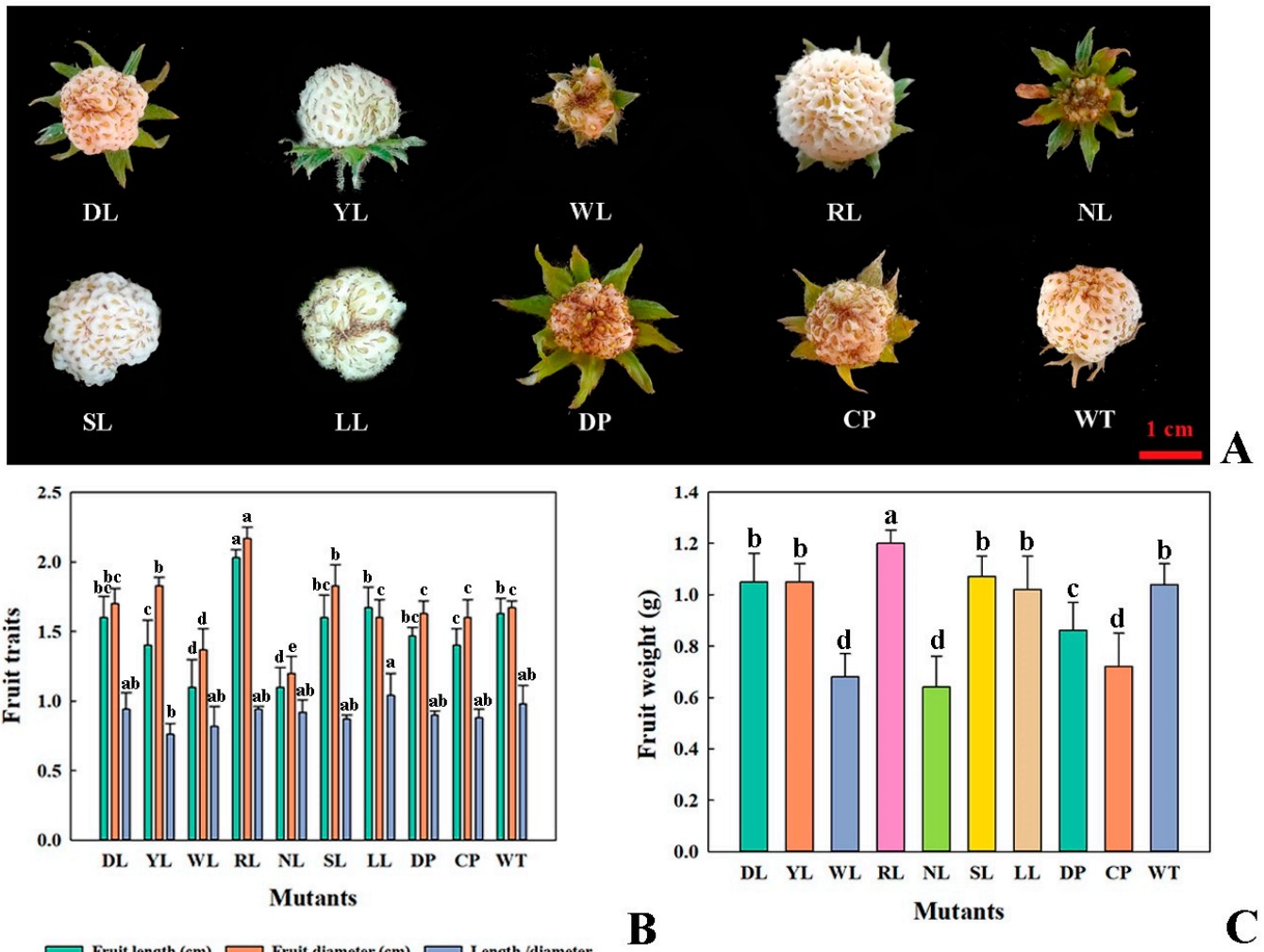

**Figure 5.** The fruit traits of 9 mutants in *F. nilgerrensis*. (**A**) Fruits of 9 mutants; (**B**) fruit length, fruit diameter, and length/diameter; (**C**) fruit weight. Abbreviated letters of mutants were described in Figure 2. Bars indicate S.E. of means, and different lowercase letters indicate significant differences between treatments according to LSD test (*p* < 0.05). Scale bar = 1 cm.

### 3.4.3. The Indexes of Photosynthesis and Fluorescence

To evaluate whether photosynthetic capacity changed or not, the photosynthetic indexes of 11 mutants were measured. The results showed that the photosynthetic indexes were significantly different between all mutants (Figure 7A,B). For example, the three photosynthetic indexes (Pn, Tr, and Gs) in the YL mutant were higher than WT. However, the value of Ci index was opposite to the above three indexes and was lower. This indicated that the net photosynthetic rate of the YL mutant was higher than that of WT.

Based on fluorescence parameters, OJIP and radar plots were plotted to compare the photosynthetic efficiency of all samples (Figure 7C,D). The OJIP curve showed that the curve of some mutants fluctuated significantly more than that of WT, especially for the YL mutant. The results of the radar chart showed that the average fluorescence parameters of the YL mutant were different from those in WT. Three parameters increased significantly in the process of photosynthetic electron transport in the YL mutant, including Vj (which reflects the light energy absorption and energy capture efficiency), dVG/dto, and dV/dto (which reflects the closing rate of the reaction center). However, the Tfm parameter (which reflects the time to reach maximum fluorescence) decreased. These indicated that the reaction center of mutants was significantly disturbed, resulting in a change in photosynthetic efficiency. The results of fluorescence and photosynthesis were consistent.

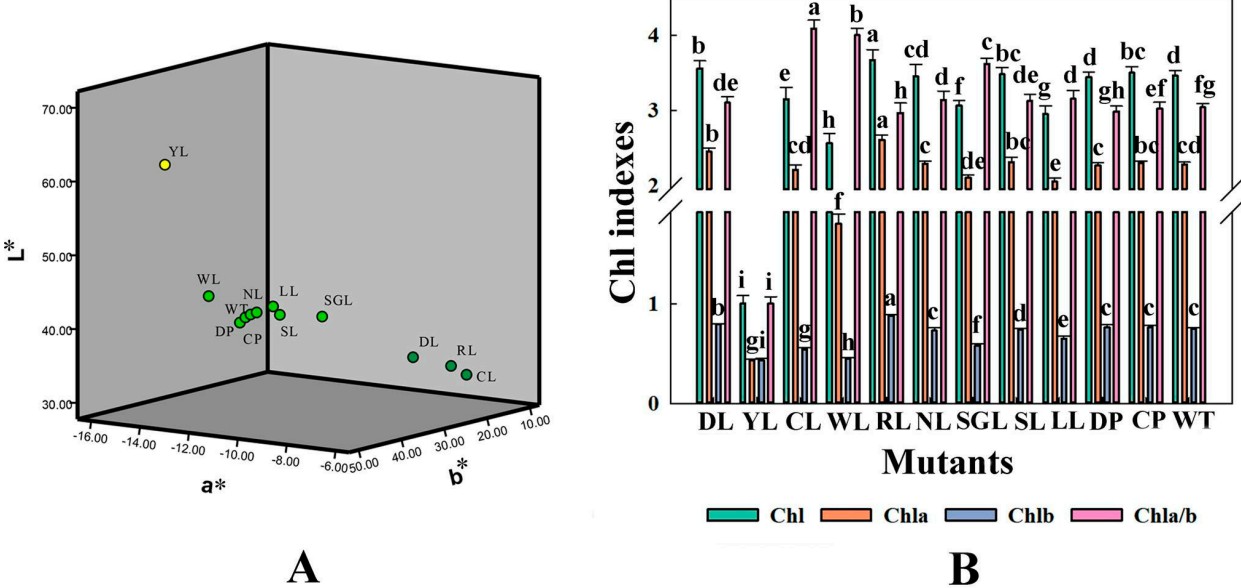

**Figure 6.** L *, a *, and b * values and chlorophyll content of 11 mutants in *F. nilgerrensis*. (**A**) Scatter plot of L *, a *, and b *; (**B**) contents of Chl, Chla, and Chlb and the ratio of Chla/b. Bars indicate S.E. of means, and different lowercase letters indicate significant differences between treatments according to LSD test ($p < 0.05$) in (**B**).

### 3.5. Cytological Observation of Mutants in F. nilgerrensis

The chromosome numbers of root-tip cells of 11 mutants and WT were observed to detect their ploidy levels. The RL, WL, and SGL mutants appeared to be tetraploids with 28 chromosomes. The other mutants processed the same number as WT, which were diploids with 14 chromosomes (Figure 8). The numbers of chromosomes in some mutants were changed via mutagenesis.

The results from flow cytometry indicated that a peak appeared at the site of around 150 in WT (Figure 9A), while the peak appeared at the site of 300 in the RL mutant (Figure 9B). Both the SGL mutant and the WL mutant showed two peaks at the sites of 150 and 300, respectively (Figure 9C,D). These indicated that their ploidy of chromosomes might have changed. The flow cytometry results were consistent with the botanical index of the transect of the two lateral leaflets/width of the central leaflet.

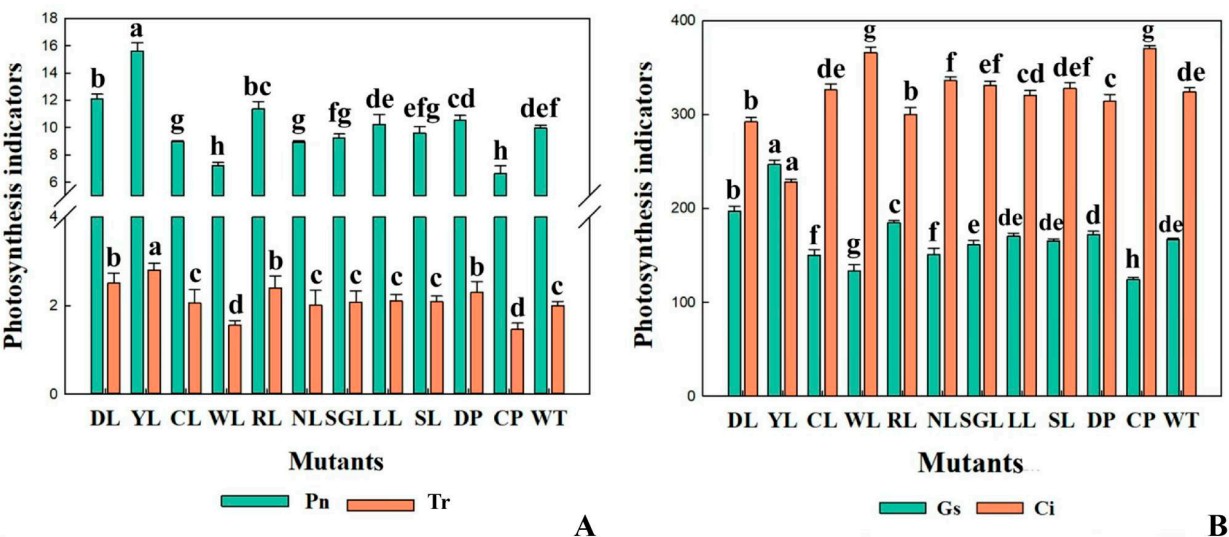

**Figure 7.** *Cont.*

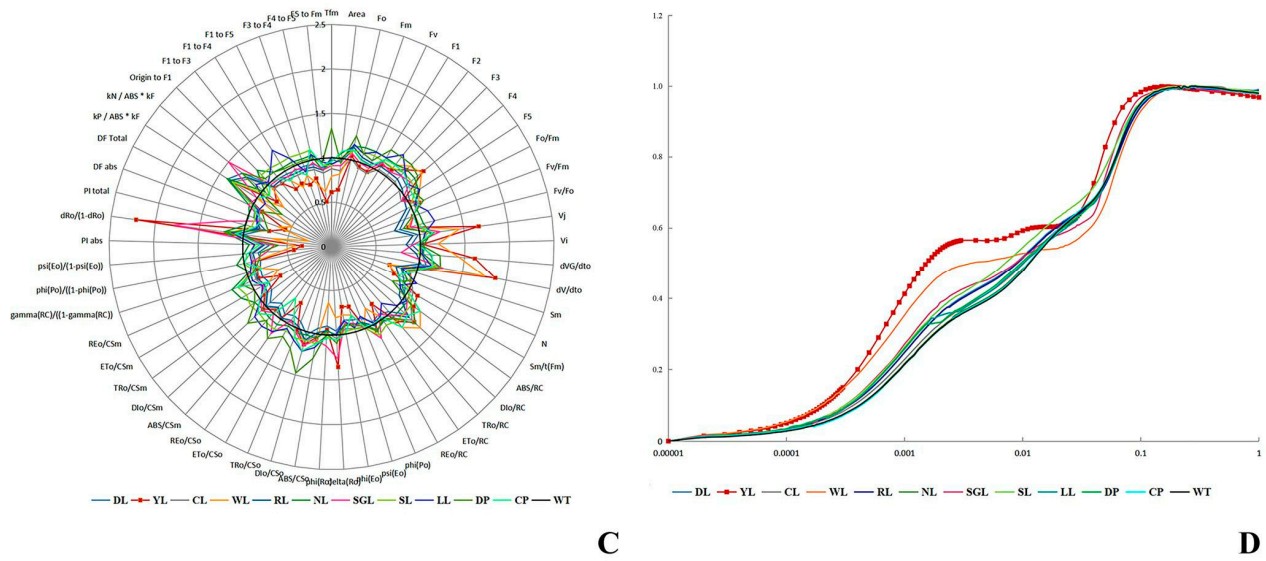

**C** **D**

**Figure 7.** Photosynthesis and fluorescence indexes of 11 mutants in *F. nilgerrensis*. (**A**) Photosynthetic rate (Pn) and transpiration rate (Tr); (**B**) stomatal conductance (Gs) and intercellular $CO_2$ (Ci); (**C**) plot of fluorescence radar; (**D**) plot of fluorescence (OJIP). Bars indicate S.E. of means, and different lowercase letters indicate significant differences between treatments according to LSD test ($p < 0.05$) in (**A**,**B**).

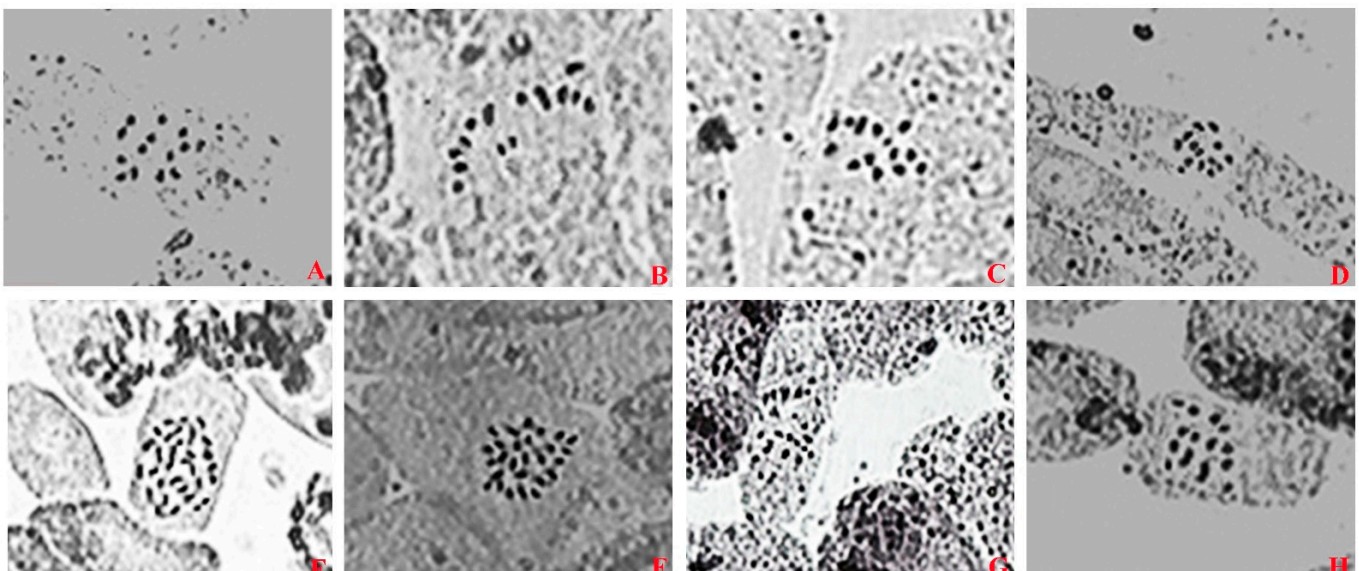

**Figure 8.** The number of chromosomes in root-tips cells of mutants in *F. nilgerrensis*. (**A**) YL: 2n = 2$x$ = 14; (**B**) CL: 2n = 2$x$ = 14; (**C**) NL: 2n = 2$x$ = 14; (**D**) SL: 2n = 2$x$ = 14; (**E**) RL: 2n = 4$x$ = 28; (**F**) WL: 2n = 4$x$ = 28; (**G**) SGL: 2n = 4$x$ = 28; (**H**) WT: 2n = 2$x$ = 14.

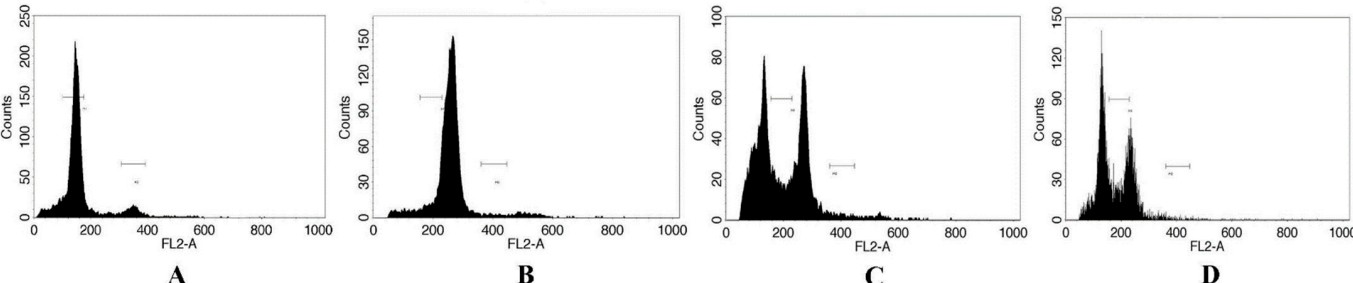

**Figure 9.** Determination of ploidy levels of several mutants in *F. nilgerrensis* by flow cytometry. (**A**) WT; (**B**) RL; (**C**) WL; (**D**) SGL.

## 4. Discussion

### 4.1. Effects of EMS Concentration and Treatment Time on Apical Meristem Mutagenesis in F. nilgerrensis

Different plant materials showed different sensitivities to EMS. The EMS treatment of seeds was the most common approach and could easily obtain a large number of mutants with a simple operation. The optimum EMS dosage for carrot and cabbage seeds was below 1%, with 18.51% and 29.88% mutation rates, respectively [17,35], but was over 1% for pepper and cucumber seeds, with 28.6% and 12.49% mutation rates, respectively [36,37]. It was reported that EMS dosage below 1% could induce the mutation of woodland strawberry seeds [25,26]. The mutant rate was high and satisfactory through the seed mutagenesis. However, screening for stably inherited mutation is usually performed in the $M_2$ generation. In recent years, other materials have also been used for EMS mutagenesis. The homozygous microspores in Chinese cabbage were treated with 0.12% EMS for 20 min, with a 2% mutation rate, and homozygous mutant plants could be screened from $M_1$ lines [38].

In this experiment, apical meristems were treated with 0.6% EMS for 6 h to develop a mutant library. A total of 86 mutants were obtained in $M_1$ regeneration, with a 17% mutation rate, which was higher than that of microspore mutagenesis and lower than that of seed mutagenesis [38,39]. Furthermore, it was observed that among the 86 mutant lines, 80 and 43 lines have already bloomed and fruited, respectively, which will have important significance for the functional genomics research and breeding program of *F. nilgerrensis*. Therefore, apical meristem mutagenesis combining the advantages of seed and microspore mutagenesis was efficient for obtaining mutants from $M_1$ lines with a relatively high mutant rate.

### 4.2. Prospect and Utilization of F. nilgerrensis Mutants

EMS is an alkylating agent and may produce two types of mutation: conversion of G:C to A:T and substitution of bases [22]. Through EMS mutagenesis, various traits, such as leaf color [40], leaf shape [26], leaf number [41], leaf size, plant height [42], flowering time [43,44], flower color [36], flower size [45], etc., could be changed. The leaf color mutants obtained in this experiment were the most abundant type of mutants, accounting for 17.43% of the total, and included yellow leaf, mottled leaf, albino leaf, and dark green leaf mutants. These mutations have a wide range of types, which are good materials for the research of gene functions of photosynthesis mechanism, chloroplast development, and chlorophyll metabolism. At the same time, mutants of leaf size and dwarf mutants were obtained via mutagenesis, accounting for 27.06% and 30.59%, respectively. Some special mutants were also obtained in this study, including the mutants with curled leaf, single leaflet, and wrinkled leaf, which have been reported previously in other crops [36,37,46]. A novel clumped plant mutant, which has not been reported previously and will provide the basis for gene function study of related traits, was also obtained.

The tetraploid mutants obtained using EMS in this experiment may be easier to cross with octoploid cultivars than diploid wild species in order to introduce the excellent characteristics of *F. nilgerrensis* into cultivated strawberries, which could provide an effective

method of polyploidy breeding in strawberries [47]. The occurrence of tetraploid, an unanticipated outcome, was most likely induced by EMS mutagenesis. In our experiment, many apical meristems of *F. nilgerrensis* were cultured as controls, and the ploidy mutants were not found. However, there was a report on several tetraploid plantlets that appeared during in vitro culture and genetic transformation of *F. vesca* "Hawaii 4" [30]. Regardless, these mutants obtained in our experiment were important materials for breeding and functional genome research of *F. nilgerrensis*.

## 5. Conclusions

EMS was used to establish a mutant library with 86 mutants of *F. nilgerrensis*. The treatment of 0.6% EMS for 6 h was the most suitable dosage for its mutation. The mutants of plant height and posture, leaf color, leaf shape, leaflet number, leaf size, flower size, and petal number were obtained and identified by morphology, physiology, and cytology, which provided important germplasm resources for the functional genome study of *F. nilgerrensis*.

**Supplementary Materials:** The following are available online at https://www.mdpi.com/article/10.3390/horticulturae8111061/s1: Figure S1. Mutant library of *F. nilgerrensis*. (a) Dark green leaf; (b) yellow-green leaf; (c) yellow leaf; (d–f) mottled yellow leaf; (g) albino leaf; (h) curled leaf; (i) wrinkled leaf; (j) round leaf; (k) narrow leaf; (l) single leaflet; (m) small leaf; (n) dwarf plant; (o) clumped plant; (p) increased petal number; (q) large flower; (r) small flower; (s) deformed flower; (t) WT. Figure S2. The leaf color and morphology phenotypes of mutants in *F. nilgerrensis*. (A) Dark green leaf; (B,C) yellow-green leaf; (D) yellow leaf; (E–H) mottled yellow leaf; (I) albino leaf; (J) curled leaf; (K) wrinkled leaf; (L) round leaf; (M) narrow leaf; (N) single leaflet; (O) large leaf; (P–S) small leaf; (T) WT. Figure S3. The flower color and morphological changes of several mutants in *F. nilgerrensis*. (A) Increased petal number; (B) large flower; (C) small flower; (D) WT. Table S1. Botanical traits of 611 surviving individuals in *F. nilgerrensis*. Table S2. Statistics on flower characteristics of 10 mutants in *F. nilgerrensis*. Table S3. Statistics on fruit characteristics of nine mutants in *F. nilgerrensis*. Table S4. Determination of L*, a*, and b* values of 11 mutant leaves in *F. nilgerrensis*.

**Author Contributions:** S.J.: conceptualization, methodology, formal analysis, investigation, fata curation, writing—original draft, visualization; M.W.: investigation, visualization. C.Z.: investigation, data curation; Y.C.: investigation, data curation; Z.C.: investigation, data curation; J.Z.: writing–review and editing, supervision; Y.Z.: writing–review and editing, supervision; L.X.: methodology, formal analysis, writing–review and editing, supervision; J.L.: methodology, data curation, visualization, validation, writing–review and editing, project administration, funding acquisition. All authors have read and agreed to the published version of the manuscript.

**Funding:** This work was supported by the Liaoning Provincial Science and Technology Project of 'Jiebangguashuai' (No. 2022JH1/10400016) and Shenyang Academician and Expert Workstation Project (2022-15).

**Institutional Review Board Statement:** Not applicable.

**Informed Consent Statement:** Not applicable.

**Data Availability Statement:** The data presented in this study are available in the article.

**Conflicts of Interest:** The authors declare no conflict of interest.

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
