# Peer review of "Establishment of a Mutant Library of Fragaria nilgerrensis Schlechtendal ex J. Gay via EMS Mutagenesis"

_horticulturae, doi:10.3390/horticulturae8111061_

Round 1

Reviewer 1 Report

I believe a better title for this work could be: “Establishment of a Mutant Library of Fragaria nilgerrensis via EMS Mutagenesis”

The aim of this work was to study the apical meristem of F. nilgerrensis’ response to EMS treatment, screening for suitable mutagenesis conditions and the construction of a mutant library.  

The introduction is concise and to the point. The methodology is well explained, the results and discussion are well-written and supported by the data obtained in the experiment. I have no concerns about this study, and I recommend its acceptance. I enjoyed reading the manuscript. Although English usage must be polished throughout the whole manuscript. For a better understanding of the text, abbreviations should be well defined using them freely. Apply italics to words such as “in vitro” and scientific names.

Reviewer 2 Report

The current paper by Jiang et. al. titled ‘Establish a Mutant Library of Fragaria nilgerrensis via EMS Mutagenesis’ has tried to focus on creating EMS based mutant library of F. nilgerrensis and subsequent functional characterization of genes. The study might be an important contribution to this area. The following few minor points need to be addressed by the authors to further improve this manuscript. 

·       Revise the title of the paper ' Establishment of Fragaria nilgerrensis mutant library via EMS Mutagenesis.   

·       Authors are suggested to replace the existing Figure 1 with clear images. 

·       Authors are suggested to rewrite the statement made in line no. 250-51 for better clarity to the readers. 

·       Please replace the existing Figure 6 with a better-quality image. 

·       Please replace existing Figures 7C and 7D with a clear and high-quality image. 

·       Check the legend of Table S1. Also, mention what does code no. in the first column of the table representing?

Reviewer 3 Report

The manuscript is well written and the experiments and results are well performed and presented. This work is of great interest as a resource for functional genomics studies in F. nilgerrensis and will also be useful for the rest of Fragaria. A good job has been carried out in the study and classification of the obtained mutants, which will also be very useful to select the mutants of interest in each case, having identified some promising mutants.

I only have a few minor comments to be reviewed by the authors:

-Please, check for extra spaces between the words, ";" followed by capital letters and so on throughout the manuscript.

-Just to be sure, in figure 2, is it the same scale in all the images? If not, the scale corresponding to 5 cm must be in each image to be able to compare correctly.

-I do not understand the values that are indicated in line 237 regarding fig 3 E and table 3, could you clarify this?

-You say on line 274 that no other changes in the fruit have been found apart from the size and weight of the fruit; does that mean that other characters were measured or are you referring only to color and fragrance? It would be clearer if the other parameters measured are specified again in results.

Please, in line 267 do not use seeds to refer to achenes, just call them achenes! And, if you want, in any case, add “(true fruit)”.

Reviewer 4 Report

Reviewer comments:

The authors present a well written and structured manuscript. The manuscript needs minor revision such as minor grammatical errors.

Figure1 is low quality

Figure 7C, Figure 9 are difficult to read.
